# A Review of the Association Between Dietary Intake and Brain Iron Levels in Older Adults: Preliminary Findings and Future Directions

**DOI:** 10.3390/nu16234193

**Published:** 2024-12-04

**Authors:** Valentinos Zachariou, Christopher E. Bauer, Colleen Pappas, Brian T. Gold

**Affiliations:** 1Department of Behavioral Science, College of Medicine, University of Kentucky, Lexington, KY 40536, USA; vzachari@uky.edu; 2Department of Neuroscience, College of Medicine, University of Kentucky, Lexington, KY 40536, USA; 3Sanders-Brown Center on Aging, University of Kentucky, Lexington, KY 40536, USA; 4Magnetic Resonance Imaging and Spectroscopy Center, University of Kentucky, Lexington, KY 40504, USA

**Keywords:** longitudinal, brain iron, aging, QSM, nutritional intake, cognitive performance

## Abstract

**Background/Objectives:** Non-heme iron is essential for critical neuronal functions such as ATP generation, synaptogenesis, neurotransmitter synthesis, and myelin formation. However, as non-heme iron accumulates with age, excessive levels can contribute to oxidative stress, potentially disrupting neuronal integrity and contributing to cognitive decline. Despite growing evidence linking high brain iron with poorer cognitive performance, there are currently no proven methods to reduce brain iron accumulation in aging or to protect cognitive function from iron’s negative effects. Recent studies suggest that nutrition may influence brain iron levels, though the evidence remains limited and mixed. **Methods:** In this review, we explore recent findings, including our own cross-sectional and longitudinal studies, to evaluate the potential effectiveness of healthy diets and specific nutrients in mitigating brain iron accumulation during aging. We also briefly assess the roles of age and gender as factors in the relationship between dietary factors and brain iron load. **Results:** The limited findings in the literature indicate that dietary choices may impact brain iron levels. In particular, nutrients such as vitamins, antioxidants, iron-chelators, and polyunsaturated fatty acids may slow brain iron accumulation in older adults. **Conclusions:** Our review highlights the multiple gaps in current knowledge and underscores a critical need for additional research on this important topic.

## 1. Introduction

Non-heme iron is essential for proper brain functioning, contributing to cellular processes such as mitochondrial adenosine triphosphate (ATP) production, synapse formation, neurotransmitter synthesis, and myelination [1,2,3,4]. However, non-heme iron is also an oxidizing agent and in excess can induce the formation of reactive oxygen species (ROS) [5]. ROS can lead to neurotoxicity and neurodegeneration due to ferroptosis [6] by interacting adversely with cellular structures [1,5,7,8,9,10,11,12].

To counter these oxidative risks, healthy brain cells store non-heme iron within ferritin complexes with mitochondria regulating its release to support cellular functions [13,14]. However, aging has been linked with excess non-heme iron accumulation in both cortical and subcortical brain regions [11,15,16,17]. Age-related alterations to cellular functions appear to impair this iron storage process, giving rise to an intracellular buildup of non-heme iron, which heightens oxidative stress and has been associated with declines in cognitive function [8,10,11,15,18,19,20,21,22].

Currently, no strategies exist to mitigate brain iron accumulation in aging or alleviate its adverse effects on cognition. However, an increasing number of studies [23,24,25,26] suggest that nutrition may play a role in regulating brain iron accumulation. For example, our recent cross-sectional and longitudinal studies suggest that specific nutrients are associated with lower brain iron accumulation in cognitively normal older adults [23,26]. The purpose of this review is to summarize the existing literature linking dietary factors to brain iron concentration in human aging and provide recommendations for future research in this field.

## 2. Scope of Review

We conducted an integrative literature review using PubMed, focusing on three categories of search terms. For age-related terms, we used ‘age’, ‘aging’, and ‘older adults’; for brain iron, we used ‘QSM’, ‘quantitative susceptibility mapping’, ‘magnetic susceptibility’, ‘R2*’, ‘relaxometry’, ‘brain iron’, and ‘SWI’; and for nutrition-based terms, we used ‘diet’, ‘dietary’, ‘nutrition’, and ‘lifestyle’. We systematically combined one term from each category (age, brain iron, and nutrition/diet) using the AND operator to generate all possible permutations. These search queries yielded 188 unique publications. We then repeated the search, adding ‘clinical trial’ as an additional term, which produced seven more results, for a total of 195 unique publications.

We then applied the following inclusion criteria to these studies: (1) published in an English-language journal up to September 2024; (2) reported in vivo MRI scans of the human brain; (3) empirical research (excluding reviews); (4) included cognitively healthy human participants; and (5) examined the relationship between brain iron—assessed using QSM, SWI, R2*, or similar modalities—and nutrition. Of the 195 studies initially identified, only four met all the inclusion criteria for our review.

## 3. Nutrition and Brain Iron: Insights from Neuroimaging Research

The study by Jesper Hagemeier and colleagues in 2015 [25] appears to be the first to explore the impact of dietary factors on brain iron levels in humans. The study examined associations between dietary factors, assessed via a diet questionnaire [27], and brain iron concentration, measured using susceptibility-weighted neuroimaging (SWI). SWI-based iron concentrations were evaluated in several deep gray matter (DGM) structures, including the bilateral caudate, putamen, thalamus, pulvinar, hippocampus, amygdala, red nucleus, and substantia nigra. The dietary factors assessed included iron and calcium supplement intake (present or absent, consumed >50% of the time over the last three years), total dairy intake (including whole milk, low-fat milk, nonfat milk, sliced cheese, whole cheese, yogurt, ice cream, milkshakes, sour cream, cream, and other dairy products), total vegetable intake (servings per week), and total red meat intake (including beef and beef products, lamb, venison, and other red meats).

Results from the Hagemeier et al. [25] study revealed that participants who took iron supplements had higher SWI-based iron concentrations in the total DGM. Additionally, higher iron concentrations were observed in vegetarians compared to non-vegetarians, thus linking dietary patterns to brain iron load. Importantly, significant gender differences emerged in the associations between dietary intake and brain iron levels. In men, higher dairy and vegetable intake was associated with increased iron concentrations in the thalamus, pulvinar nucleus, and red nucleus, while red meat intake was linked to lower iron concentrations in the hippocampus. In contrast, among women, vegetable intake was associated with lower SWI-based iron concentrations in the thalamus and pulvinar, whereas red meat intake showed a trend toward higher iron concentration in the thalamus.

The strengths of the Hagemeier et al. [25] study include being the first of its kind and having a relatively large sample size of 190 participants, which allowed for a meaningful consideration of gender differences. This study also had several limitations. First, SWI-based phase analyses, used as a proxy for brain iron concentration, are not ideal. Although sensitive to changes in iron, these analyses can also be influenced by calcium phosphates and deoxygenated blood [28]. Additionally, SWI-based phase analyses cannot differentiate between various forms of iron or their containment structures (such as hemosiderin and ferritin), despite their differing biological impacts [29]. Changes in phase, which reflect the tissue’s magnetic properties, might also be due to alterations in the chemical form of iron (e.g., from nonmagnetic oxyhemoglobin to highly magnetic transferrin and deoxyheme) rather than total iron concentration [30]. Lastly, SWI-based phase analyses, as well as relaxometry-based susceptibility measures of brain iron concentration, like T2* and R2*, can be influenced by background field inhomogeneities unrelated to iron content [31,32], and susceptibility values may differ according to the magnetic field strength and to the orientation of the head in the MRI scanner [33].

A very important factor to consider in this and other studies we summarize in this review relates to the age of participant samples. Hagemeier et al. [24] focused on young to middle-aged adults, and the relationship between brain iron and cognition varies across the lifespan. Better cognitive ability is associated with higher brain iron levels in late adolescence and young adulthood [34], while in older adults, the opposite trend is observed [8,10,11,15,18,19,20,21,22]. Consequently, the associations between brain iron and nutrition may differ in older individuals, who typically exhibit higher brain iron concentrations.

Finally, the nutritional information considered in this study was relatively broad (total dairy intake, total vegetable intake, and total red meat intake), and it was primarily motivated by factors related to iron absorption in the body. Although some studies suggest an association between iron absorption, reflected in markers of iron storage and transport in the blood, and brain iron accumulation in older adults [24,35], the underlying reasons for this association remain unclear.

Chronologically, our group was the next to investigate associations between dietary factors and brain iron in both cross-sectional [23] and longitudinal [26] studies. Our research included 73 older adults between the ages of 61 and 86 years old and focused on specific nutrients, namely vitamins, antioxidants, flavonoids, iron-chelating nutrients, and polyunsaturated fatty acids (PUFAs). Dietary intake was evaluated using the 92-item “Newly Developed Antioxidant Nutrient Questionnaire” [36]. Responses from each questionnaire were analyzed using custom-developed software, which converts daily food consumption into milligrams (mg) of 122 nutrients per day for each participant. A targeted literature review was subsequently conducted to identify a subset of these 122 nutrients most pertinent to our study objectives based on prior evidence indicating the nutrient’s ability to cross the blood–brain barrier, reduce oxidative stress, chelate brain iron and/or protect against ferroptosis, which is a cell death process driven by the interaction of reactive oxygen species with iron [37,38].

A significant methodological improvement in our work was the use of quantitative susceptibility mapping (QSM) to assess brain iron concentrations. QSM offers several advantages over more traditional SWI-based phase and relaxometry-based measures [28,39,40], including being a more representative measure of intrinsic tissue susceptibility, its independence from magnetic field strength, and its reduced susceptibility to background field inhomogeneities or head positioning in the MRI scanner [33,41]. However, a limitation of QSM, similar to SWI-based phase and relaxometry-based measures, is its inability to differentiate between non-heme and heme iron bound to deoxygenated hemoglobin in blood [31]. Despite this limitation, a previous study from our group [21] found no significant associations between cerebral blood volume in lobar gray matter and QSM values extracted from the same regions after removing outlier QSM values with intensities comparable to those observed in large draining veins. This suggests that following outlier removal, any potential contribution of cerebral blood volume to the QSM signal is small.

Our study’s cross-sectional analysis showed that a greater consumption of antioxidants, vitamins, iron-chelating nutrients, and PUFAs correlated with better working memory performance (Figure 1) and lower levels of brain iron (Figure 2). Consistent with these findings, our longitudinal results [26] revealed that brain iron accumulation, observed in multiple cortical and subcortical brain regions over a 2.5- to 3-year period, was reduced as a function of baseline dietary consumption of these nutrients (Figure 3). Specifically, we found that nutrients typically found in fruits and vegetables were linked to reduced iron accumulation in the basal ganglia, particularly the pallidum. Antioxidants, iron chelators, and PUFAs had a comparable effect in cortical regions, including the frontal cortex and hippocampus. Our combined cross-sectional and longitudinal results suggest that brain iron buildup during aging may not be immutable but modifiable through dietary habits.

However, several important limitations of our previous studies must be considered. First, in our previous longitudinal work, we did not examine whether nutrition moderates the association between cognitive outcomes and brain iron accumulation, leaving open the question of whether the negative associations we found between nutrition and iron accumulation have cognitive significance. Second, both our cross-sectional and longitudinal studies are observational, limiting the ability to draw causal inferences that would be possible through randomized clinical trials. Lastly, whether the effects we observed between nutrition and brain iron were driven by the specific nutrients we investigated, or if they reflect a broader pattern of healthier dietary habits among participants with higher intake of antioxidants, vitamins, iron-chelating nutrients, and PUFAs, remains an open question.

The latest study to explore the relationship between brain iron and nutrition offers some further insights into this unanswered question. Gustavsson et al. [24] assessed QSM-based brain iron concentrations and accumulation within the cortex and basal ganglia in a cohort of 208 healthy volunteers aged 20–79 both at baseline and at a follow-up approximately three years later. Diet was evaluated using a semi-quantitative food frequency questionnaire, which captured three unhealthy food groups (butter and margarine, red meat products, pastries and sweets) and 10 healthy food groups (berries, whole grains, beans, nuts, green leafy vegetables, other vegetables, cooking oils such as olive and rapeseed, cheese, fish, and poultry). Based on this information, the authors calculated a Mediterranean-DASH Intervention for Neurodegenerative Delay (MIND) score for each participant with higher scores indicating healthier dietary habits [42].

The study revealed that a healthier diet was associated with increased iron in the basal ganglia in younger adults, while older adults showed a trend in the opposite direction, although this finding was not statistically significant. These observations align somewhat with our previous work, which demonstrated that the consumption of specific “healthy” nutrients was related to lower brain iron content in an older population. However, the lack of significance in this finding may suggest that adhering to a healthy diet alone may not be sufficient to reduce brain iron accumulation in older individuals, where iron overload can have detrimental effects. More importantly, given the challenges in interpreting null results, these findings highlight the need for additional research to evaluate associations between overall dietary habits and brain iron load and accumulation.

The study of Gustavsson et al. [24] shares similar limitations with our work. First, being an observational study, it does not allow for causal inferences that could be drawn from randomized clinical trials. Second, the study did not assess associations between nutrition, brain iron accumulation, and cognition. As a result, it remains unclear whether the effects of healthier diets on brain iron accumulation—whether positive or negative—carry cognitive significance.

## 4. Conclusions from the Neuroimaging Findings

Growing evidence highlights the detrimental effects of iron accumulation on brain integrity and cognition in older adults [8,10,11,15,18,19,20,43,44], yet the factors influencing brain iron content and accumulation remain poorly understood. For example, we could find only four published studies to date that have explored the impact of dietary factors on brain iron concentration and accumulation, yielding mixed results. Nonetheless, the limited findings suggest that dietary choices can influence brain iron levels. Specifically, the consumption of vitamins, antioxidants, iron-chelating nutrients, and polyunsaturated fatty acids may slow brain iron accumulation in older individuals [23,26].

Given the limited understanding of the impact of dietary factors on brain iron accumulation, there is a pressing need for further research in this area. We urge future studies to examine associations between brain iron load and nutrition during normal aging while also considering the influence of biological factors such as age and gender as well as lifestyle factors like smoking, alcohol consumption, and physical activity, along with their potential interactions.

## 5. Proposed Future Directions

In this section, we offer specific recommendations for future research directions. First, healthy diets, such as the Mediterranean and DASH (dietary approaches to stop hypertension) diets, could be directly compared with less healthy diets such as the Western diet, to identify which specific component nutrients may be most effective in reducing, versus increasing, brain iron accumulation.

Second, future clinical trials are needed to assess the effects of specific nutrients in lowering brain iron while controlling for overall dietary habits. Findings from animal research should be considered in determining the specific nutrients to be tested. For example, evidence from animal models suggest that specific nutrients, such as vitamin E, acetyl-l-carnitine and α-lipoic acid, can slow or even reverse the age-related iron accumulation and antioxidant depletion in the rat cerebral cortex [45,46]. Additionally, hepcidin, a regulator of iron homeostasis [47], could also play a role, as nutritional factors that influence hepcidin levels may affect iron accumulation. The impact of dietary factors should also be evaluated in relation to both iron accumulation and cognition. Specifically, determining whether dietary factors can lessen the negative impact of brain iron accumulation on cognitive function.

Future studies should also be conducted to better understand the relationship between dietary intake of iron and brain iron levels. For example, it has been hypothesized that excessive dietary intake of iron may contribute to increased brain iron accumulation over time [48]. In this context, iron absorption modulators may be of interest. Phytates (found in grains and legumes), polyphenols (in tea and coffee), and calcium can inhibit iron absorption, making the regulation of iron metabolism a potentially important factor in reducing brain iron accumulation in normal aging.

Furthermore, the role of blood–brain-barrier integrity and chronic inflammation in relation to brain iron accumulation warrants further investigation. For example, inflammation can disrupt iron metabolism, promoting iron accumulation [49], and dietary factors that modulate inflammatory processes [50] may indirectly help reduce brain iron accumulation.

Additionally, the relationship between brain iron accumulation, its effects on cognitive functions, and mediating factors in the context of iron overload disorders like hemochromatosis remains understudied and calls for further research. For example, individuals with a high genetic risk for hemochromatosis exhibit elevated brain iron concentrations, as determined by MRI (T2-weighted and T2*-based signal intensities), in the basal ganglia, thalamus, red nucleus, and cerebellum [51]. However, it remains unclear whether this link between hemochromatosis risk and increased brain iron concentrations affects cognitive or motor functions and whether factors that appear to mediate more typical age-related brain iron accumulation, such as nutrition, might also mitigate hemochromatosis.

Finally, we encourage the publication of negative findings of well-powered studies in this field to avoid confirmation bias. Insights from continued research in the area of nutrition and brain iron load could inform future intervention trials aimed at mitigating age-related brain iron accumulation through nutritional and lifestyle modifications.

## Figures and Tables

**Figure 1 nutrients-16-04193-f001:**
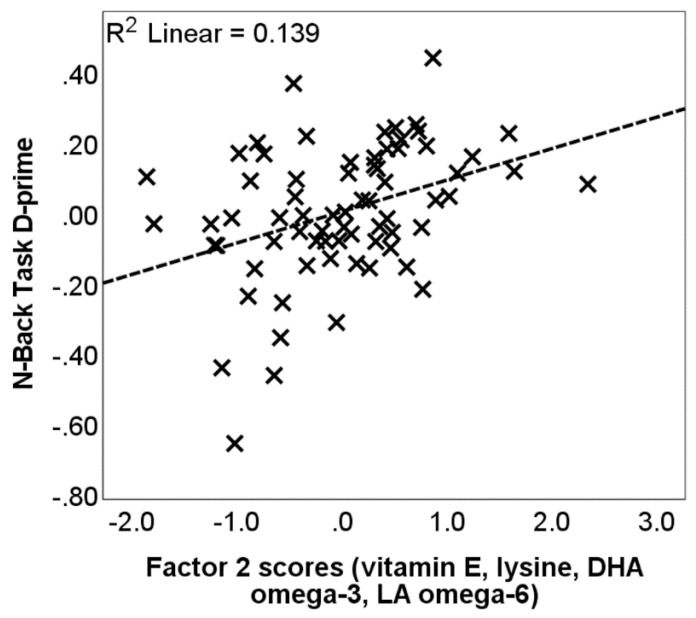
The relationship between dietary intake and working memory performance (adapted from Figure 3 in [23]). The scatter plot shows the factor 2 scores (composed of vitamin E, lysine, DHA omega-3 and LA omega-6 nutrients) on the x-axis against N-back task performance (log D-prime, averaged across the 1-Back and 2-Back conditions) on the y-axis. Values are standardized residuals after controlling for age, gender, years of education, PASE score and alcohol drinking frequency (drinks/day). The dashed line represents the linear best fit.

**Figure 2 nutrients-16-04193-f002:**
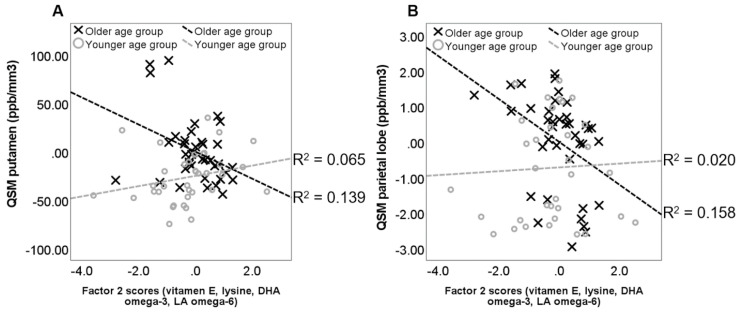
Interaction between age and dietary intake on regional QSM values (adapted from Figure 4 in [23]). The scatter plot plots QSM values (iron concentration in ppb/mm^3^) from (**A**) the putamen and (**B**) the parietal lobe, against factor 2 scores. Separate regression lines are depicted for the older (X) and younger (O) age groups (median age = 70.22 years). Values are standardized residuals after controlling for gender and years of education, PASE score and alcohol drinking frequency (drinks/day). The dashed lines represent the linear best fit.

**Figure 3 nutrients-16-04193-f003:**
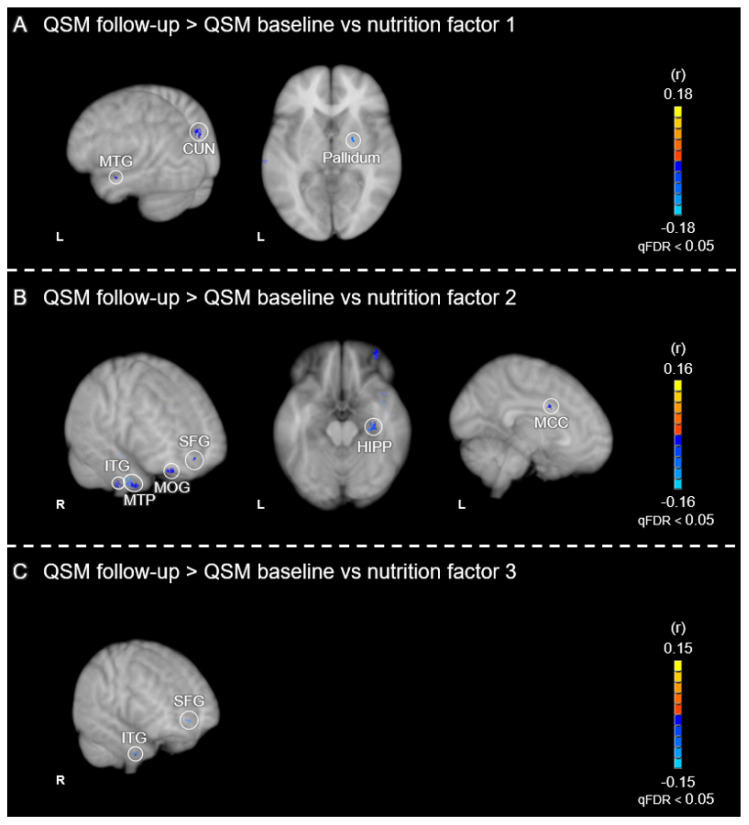
Voxel-wise correlations between longitudinal change in QSM values and nutrition factor scores (adapted from Figure 3 in [26]). The correlations depict the relationships between the MRI contrast of (QSM follow-up > QSM baseline) vs. (**A**) nutrition factor 1 (vitamin C, quercetin, β-carotene and β-cryptoxanthin) (**B**), nutrition factor 2 (vitamin E, lysine, DHA omega-3 and LA omega-6) and (**C**) nutrition factor 3 (epigallocatechin 3-gallate). Positive correlations are shown in warm (orange to yellow) colors and negative correlations in cool (blue to cyan) colors, which are expressed as Pearson’s r values. Results are overlaid on the 1 mm, MNI152 template provided with FSL, rendered as a 3D volume. Notes: L = left hemisphere; R = right hemisphere; HIPP = hippocampus; SFG = superior frontal gyrus; CUN = cuneus; MTP = medial temporal pole; ITG = inferior temporal gyrus; MTG = middle temporal gyrus; MOG = middle orbital gyrus; MCC = middle cingulate cortex.

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
