# Peer review of "A Review of the Association Between Dietary Intake and Brain Iron Levels in Older Adults: Preliminary Findings and Future Directions"

_nutrients, 2024, doi:10.3390/nu16234193_

Round 1
Reviewer 1 Report
Comments and Suggestions for Authors
This is a well-written and informative review. However, since the authors restricted the inclusion criteria to very specific terms, for example MRI scans, it turned out that only 4 out of 195 papers matched their search criteria with 2 out of 4 studies being from their own group. However, the authors analyse and describe well the strengths and weaknesses/limitations of each study illustrating them with some graphs and images from their own 2-3 publications on the subject. They also give good recommendations for future research direction.
Author Response
Comment 1: This is a well-written and informative review. However, since the authors restricted the inclusion criteria to very specific terms, for example MRI scans, it turned out that only 4 out of 195 papers matched their search criteria with 2 out of 4 studies being from their own group. However, the authors analyse and describe well the strengths and weaknesses/limitations of each study illustrating them with some graphs and images from their own 2-3 publications on the subject. They also give good recommendations for future research direction.
Response: Thank you for your thoughtful and encouraging feedback. We greatly appreciate your kind words. We are grateful for the opportunity to contribute to this important area of research.
Reviewer 2 Report
Comments and Suggestions for Authors
The manuscript by Zachariu et al., "The Association between Dietary-Intake and Brain Iron Levels 2 in Older Adults: Preliminary Findings and Future Directions," summarizes recent studies and authors' own research to evaluate the effectiveness of dietary intervention in brain iron accumulation. The manuscript addresses a significant problem. I have a minor suggestion. The authors summarized critical factors affecting brain iron accumulation in the last section of Proposed Future Directions. It would be beneficial to supplement a Proposed Future Directions section with a graphical scheme describing the current knowledge of the factors affecting brain iron accumulation from animal and human studies.
Author Response
Comment 1: The manuscript by Zachariu et al., "The Association between Dietary-Intake and Brain Iron Levels 2 in Older Adults: Preliminary Findings and Future Directions," summarizes recent studies and authors' own research to evaluate the effectiveness of dietary intervention in brain iron accumulation. The manuscript addresses a significant problem. I have a minor suggestion. The authors summarized critical factors affecting brain iron accumulation in the last section of Proposed Future Directions. It would be beneficial to supplement a Proposed Future Directions section with a graphical scheme describing the current knowledge of the factors affecting brain iron accumulation from animal and human studies.
Response: Thank you for bringing this to our attention. Figure 4 has been added to the revised version of the manuscript on page 8, under the "Proposed Future Directions" section:
Figure 4. Schematic image of a human brain composed of foods rich in Vitamin E, Lysine, DHA omega-3, LA omega-6, and Epigallocatechin 3-gallate (EGCG). These nutrients have been associated with reduced age-related brain iron accumulation in previous studies. Representative food sources include avocados (Vitamin E), cod and chickpeas (Lysine), salmon (DHA omega-3), walnuts (LA omega-6), and green tea (EGCG).
Reviewer 3 Report
Comments and Suggestions for Authors
It has been reported that brain iron levels increase with age and that they may have negative effects on neurodegenerative disorders and perhaps also on cognition. The work by Zachariou et al. tries to summarize in a review the papers that analyzed dietary regimens and brain iron levels by MRI in humans. They found only 4 papers of interest, two of which were by the author’s group, which suggested that nutrition may affect brain iron levels differently in the younger and older populations. It does not seem to add much to the field.
I am surprised that the authors did not mention if iron disorders with major accumulation or depletion of body iron have an evident effect on brain levels.
The review is very limited, being based on only 4 papers, two of which are of the same group. In addition, it seems much self-referential, showing figures only of the author’s group. Fig. 3 is the same as fig. of Zachariou et al, 2024, and fig 1 and 2 are derived from Zachariou et al, 2021
Author Response
Comment 1: It has been reported that brain iron levels increase with age and that they may have negative effects on neurodegenerative disorders and perhaps also on cognition. The work by Zachariou et al. tries to summarize in a review the papers that analyzed dietary regimens and brain iron levels by MRI in humans. They found only 4 papers of interest, two of which were by the author’s group, which suggested that nutrition may affect brain iron levels differently in the younger and older populations. It does not seem to add much to the field.
Response 1: We appreciate the reviewer’s perspective and would like to clarify that our analysis did not focus on merely four papers of interest; rather, our extensive search of the literature revealed that only four studies exist on this specific topic—factors that may mitigate age-related brain iron accumulation in humans. Our article aims to underscore the significant gap in research within this domain and highlight the urgent need for further investigation.
Comment 2: I am surprised that the authors did not mention if iron disorders with major accumulation or depletion of body iron have an evident effect on brain levels.
Response 2: We appreciate the reviewer’s comment and acknowledge the importance of understanding the relationship between systemic iron disorders and brain iron levels. However, the primary focus of our review was specifically on factors—particularly nutrition-based factors—that may mitigate age-related brain iron accumulation in generally healthy older adults. While systemic iron disorders are indeed a critical area of study, our aim was to address the narrower question of how modifiable dietary and lifestyle factors might influence brain iron levels, an area with significant gaps in the existing literature. However, we do briefly touch on the literature between systemic iron in the Proposed Future Directions Section, on pages 8-9 of the manuscript:
“Future studies should also be conducted to better understand the relationship between dietary-intake of iron and brain iron levels. For example, it has been hypothesized that excessive dietary-intake of iron may contribute to increased brain iron accumulation over time (Hare et al., 2017). In this context, iron absorption modulators may be of interest. Phytates (found in grains and legumes), polyphenols (in tea and coffee), and calcium can inhibit iron absorption, making the regulation of iron metabolism a potentially important factor in reducing brain iron accumulation in normal aging.”
We have added the following paragraph on page 9 of the revised manuscript noting that exploring the link between systemic iron disorders and brain iron is an important avenue for future research.
“Additionally, the relationship between brain iron accumulation, its effects on cognitive functions, and mediating factors in the context of iron overload disorders like hemochromatosis remains understudied and calls for further research. For example, individuals with a high genetic risk for hemochromatosis exhibit elevated brain iron concentrations, as determined by MRI (T2-weighted and T2*-based signal intensities), in the basal ganglia, thalamus, red nucleus, and cerebellum (Loughnan et al., 2022). However, it remains unclear whether this link between hemochromatosis risk and increased brain iron concentrations affects cognitive or motor functions and whether factors that appear to mediate more typical age-related brain iron accumulation, such as nutrition, might also mitigate hemochromatosis.”
Loughnan, R., Ahern, J., Tompkins, C., Palmer, C.E., Iversen, J., Thompson, W.K., Andreassen, O., Jernigan, T., Sugrue, L., Dale, A., Boyle, M.E.T., Fan, C.C., 2022. Association of Genetic Variant Linked to Hemochromatosis With Brain Magnetic Resonance Imaging Measures of Iron and Movement Disorders. JAMA Neurol 79, 919–928. https://doi.org/10.1001/JAMANEUROL.2022.2030
Comment 3: The review is very limited, being based on only 4 papers, two of which are of the same group. In addition, it seems much self-referential, showing figures only of the author’s group. Fig. 3 is the same as fig. of Zachariou et al, 2024, and fig 1 and 2 are derived from Zachariou et al, 2021
Response 3:
We appreciate the reviewer’s observation and would like to address the concerns raised. First, our review indeed highlights findings from a limited number of studies, reflecting the scarcity of research on nutrition-based factors that may mitigate age-related brain iron accumulation. This underscores the critical gap in the literature and the need for more investigation in this domain.
Regarding the figures, we intentionally used those from our own work, as we are certain of owning the copyright and can reproduce them without requiring additional permissions. Figures from other studies would very likely require explicit approval from the respective journals for reproduction, which could have complicated the publication process—particularly given the limited timeframe allowed for review and response to reviews for this special issue. To enhance accessibility and ensure alignment with the focus of the review, we chose to include our own figures.
Round 2
Reviewer 3 Report
Comments and Suggestions for Authors
The answers to my raised points are unsatisfactory and do not eliminate my concerns. The manuscript did not improve